# The capacity of origins to load MCM establishes replication timing patterns

**Livio Dukaj****, Nicholas Rhind** *

Department of Biochemistry and Molecular Pharmacology, University of Massachusetts Medical School Worcester, Massachusetts, United States of America

* nick.rhind@umassmed.edu

**Data Availability Statement:** The sequencing data from this study have been submitted to the NCBI Sequence Read Archive under accession number PRJNA663099.

**Funding:** This study was funded by the National Institute of General Medical Sciences <https://

## Abstract

Loading of the MCM replicative helicase at origins of replication is a highly regulated process that precedes DNA replication in all eukaryotes. The stoichiometry of MCM loaded at origins has been proposed to be a key determinant of when those origins initiate replication during S phase. Nevertheless, the genome-wide regulation of MCM loading stoichiometry and its direct effect on replication timing remain unclear. In order to investigate why some origins load more MCM than others, we perturbed MCM levels in budding yeast cells and, for the first time, directly measured MCM levels and replication timing in the same experiment. Reduction of MCM levels through degradation of Mcm4, one of the six obligate components of the MCM complex, slowed progression through S phase and increased sensitivity to replication stress. Reduction of MCM levels also led to differential loading at origins during G1, revealing origins that are sensitive to reductions in MCM and others that are not. Sensitive origins loaded less MCM under normal conditions and correlated with a weak ability to recruit the origin recognition complex (ORC). Moreover, reduction of MCM loading at specific origins of replication led to a delay in their replication during S phase. In contrast, over-expression of MCM had no effects on cell cycle progression, relative MCM levels at origins, or replication timing, suggesting that, under optimal growth conditions, cellular MCM levels are not limiting for MCM loading. Our results support a model in which the loading capacity of origins is the primary determinant of MCM stoichiometry in wild-type cells, but that stoichiometry is controlled by origins' ability to recruit ORC and compete for MCM when MCM becomes limiting.

## Author summary

The coordinated replication of complex eukaryotic genomes is primarily regulated at the level of replication initiation. Therefore, how initiation is regulated in time and space is an active field of research. A leading hypothesis is that the stoichiometry of MCM, the replicative helicase, at replication origins is a significant determinant of initiation timing. However, how MCM stoichiometry at origins is regulated is unknown. We tested two hypothetical mechanisms for the control of MCM stoichiometry: Origin capacity, in which MCM stoichiometry is constrained by the varying capacity of individual origins to

www.nigms.nih.gov> grant GM125872 to NR. The
funders had no role in study design, data collection
and analysis, decision to publish, or preparation of
the manuscript.

**Competing interests:** The authors have declared
that no competing interests exist.

load MCM, and ORC activity, in which MCM stoichiometry is regulated by the varying
activity of ORC, the MCM loader, at individual origins. We find that neither mechanism
is consistent with our data, but that a hybrid of the two mechanisms is.

## Introduction

The initiation of DNA replication is an exquisitely orchestrated and highly conserved process.
Although the molecular biochemistry of initiation at individual origins continues to be eluci-
dated in great detail [1,2], the mechanism governing the time at which different regions of the
genome replicate has remained largely elusive [3]. Nonetheless, replication timing has been
implicated in a number of important processes, including gene expression, mutation rates,
and chromatin structure [4–7]. Which molecular mechanisms underlie the replication timing
patterns, how they establish it, and what the consequences are when those mechanisms go
awry is still a matter of active research [8].

Origin licensing, the establishment of potential origins of replication in G1, is separated
from replication initiation through a variety of biochemical mechanisms, which are particu-
larly well-understood in budding yeast [1,9]. In late M and G1, the Minichromosome Mainte-
nance (MCM) complex, the heterohexameric catalytic core of the replicative helicase,
composed of Mcm2-7, is shuttled into the nucleus with the help of Cdt1 [10]. In the nucleus,
the Origin Recognition Complex (ORC), which loads MCM at origins, recognizes and binds
origins at their ARS Consensus Sequence (ACS) [11,12]. After binding of its Cdc6 subunit,
ORC recruits MCM-Cdt1 to the origins and loads MCM onto double stranded DNA
(dsDNA), at which point Cdt1 is released and a second MCM is coordinately loaded [13]. The
resulting MCM double hexamer (MCM-DH) establishes an origin that can be activated for
replication through the action of the replication kinases CDK and DDK and accessory initia-
tion factors [14].

Although the biochemical mechanisms of MCM activation appear to be uniform across ori-
gins, the timing of MCM activation varies from origin to origin. The observation that MCM
activation is regulated by limiting initiation factors [15] has led to the model that the competi-
tion between origins for these limiting initiators establishes the relative efficiency of MCM
activation, which in turn determines the average firing time for each origin. This model raises
the question of why some origins compete more efficiently for the limiting factors. Hetero-
chromatin reduces origin activation efficiency, perhaps by restricting access to limiting initia-
tors, but also by recruiting phosphatases that antagonize the required S-phase kinases [16].
Conversely, origins can by programed for early activation by recruiting limiting factors. An
early example was the demonstration that fission yeast recruits DDK to centromeres to
advance their activation [17].

Although the simplest models of origin function posit exactly one MCM-DH loaded at each
origin, many studies have reported that the levels of MCM signal at origins vary by over ten
fold [12,18–21]. Moreover, MCM stoichiometry (that is, the average MCM loading at an ori-
gin, relative to other origins) has been proposed as an important determinant of replication
timing [20–22]. In particular, if MCMs are activated stochastically, then an origin that has
higher MCM stoichiometry has a greater probability of firing and thus, on average, will fire
earlier in S phase [23]. Combining evidence for the regulation of origin activation efficiency by
*trans*-acting factors with evidence for the effect of MCM stoichiometry on origin activation
efficiency leads to a unified model in which the probability of an origin firing, and thus its

average firing time in S phase, is a product of an origin's MCM stoichiometry and the activation efficiency of that MCM, regulated by *trans*-acting factors.

There are two ways in which MCM stoichiometry could vary at origins. In the one-MCM scenario, an origin either has one or zero MCM-DH loaded. Therefore, the population-based average can range from 0 to 1, a parameter that has been referred to as competency [24–26]. In the multiple-MCM scenario, more than one MCM can be loaded at an origin and so MCM stoichiometry can range from 0 to many [20,22]. As elaborated in the Discussion, distinguishing between the one-MCM and many-MCM scenarios is an area of active research. However, the hypotheses tested here—regarding whether variable MCM stoichiometry affects replication timing and how variable MCM stoichiometry is established—are independent of whether the range of variation is from 0 to 1 or 0 to many.

Regardless of the absolute stoichiometry of MCM loading at origins, it remains unclear how relative loading dynamics are regulated at individual origins genome-wide. In order to investigate the mechanisms regulating the level of MCM loading and to determine the effect that such changes in levels have on replication timing, we performed experiments under conditions in which cellular MCM levels were either increased or decreased. We aimed to test two general models of the regulation of MCM loading: the ORC Activity model, in which the relative activity of ORC at individual origins regulates MCM occupancy, and the Origin Capacity model, in which the intrinsic capacity of each origin for MCM loading regulates MCM occupancy. Importantly, we performed both MCM ChIP-seq and replication timing assays in the same experiments, such that the density of MCM loaded in late G1 could be directly correlated to replication timing at specific origins. Furthermore, we used MNase ChIP-seq, which produces much higher resolution data than standard ChIP-seq, allowing us to localize where MCMs are loaded within origins with close to base-pair resolution. Using genome-wide MNase ChIP-seq and replication timing assays, we found that lowering cellular MCM levels caused differential loading of MCM, with MCM loading at some origins being sensitive to intermediate reductions of cellular MCM levels and loading at other origins being largely resistant. Importantly, reduction in MCM levels correlated with delays in replication timing. Conversely, increasing the MCM levels in budding yeast had no effects on MCM loading onto DNA or replication timing in S phase, indicating that MCM loading under normal conditions is saturated. These results suggest that loading of MCM at origins in G1 is a dynamic process and that relative levels of helicase at origins are dependent on the origins' abilities to recruit MCM as well as the levels of MCM pools in the cells. In particular, as described in the Discussion, our results are consistent with a hybrid model combining regulation by both ORC activity and origin capacity.

## Results

### Auxin-induced reduction in cellular MCM levels causes a dose-dependent reduction in viability and slower progression through S phase

To test the effects of reduced MCM pools on helicase loading and replication timing we employed the auxin-inducible degron (AID) system optimized for budding yeast [27,28]. In order to reduce the cellular pool of MCM, we tagged the endogenous copy of Mcm4 with the AID degron cassette (IAA17) followed by GFP at its C terminus. The MCM hexamer requires all six components for nuclear localization [29]. In addition, degradation of any single component of the hexamer causes destabilization of the rest [29]. Therefore, degradation of Mcm4 is expected to reduce the total cellular pool of MCM. As previously reported, continual exposure to increasing amounts of auxin caused a dose-dependent lethality in cells harboring degron-

tagged Mcm4 (**S2A Fig**) [27]. However, cells maintain viability under our specific experimental conditions (**S2B Fig**).

We determined the effect of lowered MCM levels on genome-wide loading and replication timing in synchronized cells. We synchronized the cells and reduced MCM levels prior to G1 in order ensure that MCM was not removed from active replisomes or pre-loaded origins. We first synchronized cells in metaphase using nocodazole, followed by release into α-factor for a G1 arrest (**S1 Fig**). Mcm4 was degraded through the addition of auxin, first for 30 minutes during nocodazole arrest, then during the majority of the α-factor arrest. Following a 30 minute period of equilibration in the absence of auxin at the G1/S boundary, samples were collected for MCM ChIP-seq analysis, and the rest of the culture was released into S phase (**Fig 1A**). Immunoblots monitoring the levels of Mcm4-IAA17-GFP revealed that Mcm4 is efficiently degraded after addition of auxin in a dose-dependent manner (**Fig 1**B). Quantification of Mcm4 levels revealed that, compared to untreated cells, 30 μM auxin reduced Mcm4 levels to 14% of the endogenous levels, whereas 500 μM auxin reduced those levels to 7%. The observation that cells released onto plates containing the replication stress drug hydroxyurea (HU) showed an auxin dose-dependent increase in sensitivity to HU suggests that lowered Mcm4 levels reduce MCM function and cell viability (**S2B Fig**). We performed flow cytometry analysis on S-phase-synchronized cells treated with 0 μM, 30 μM, and 500 μM auxin and found that degradation of Mcm4 also causes a dose-dependent delay in progression through S phase (**Figs 1C** and **S1**). Together, this data show that the AID system can be used to reduce Mcm4 levels in a dose-dependent manner and that reduction of Mcm4 causes reduced viability, slower progression through S phase, and lower resistance to replication stress.

## Reduced MCM levels cause a reduction in helicase loading at many, but not all origins of replication

Previous studies using ChIP have shown that MCM abundance at origins of replication varies throughout the genome [12,18–21]. To test how lowered MCM pools affect the dynamics of helicase loading and abundance at origins of replication, we used α-factor-arrested cells to perform ChIP-seq using a polyclonal antibody against the full MCM hexamer on micrococcal nuclease- (MNase) digested chromatin [30]. MNase digestion has the advantage, relative to the standard ChIP approach of shearing by sonication, of cleaving the chromatin into much smaller fragments. Because the MCM-DH complex protects about 70 bp of DNA, using MNase ChIP-seq, we can map MCM footprints with almost nucleotide resolution.

Fragments of the genome associated with MCM were pulled down, sequenced, and their abundance was normalized to read depth and no-immunoprecipitation input controls. Using this method, we found that MCM peaks localize to known origins of replication (**Fig 1D**), are reproducible (r = 0.97–0.99, **S3A Fig**) and correlate well with previously reported levels (r = 0.78–0.82, **S3B Fig**). Quantitation of origin peaks revealed changes to MCM loading throughout the genome in response to reduction of MCM levels (**Fig 2** and **S1 Table**). Many origins exhibited a reduction in MCM abundance in auxin-treated cells, as seen by a shift away from the diagonal line in **Fig 2A**. The level of reduction was dose-dependent, as cells treated with 500 μM auxin showed greater reduction in MCM abundance than those treated with 30 μM auxin. Plotting the percent of MCM abundance that is lost in response to auxin treatment showed that some origins lose most of their MCM signal after 30 μM auxin treatment, while others are resistant to the changes in MCM levels even after 500 μM auxin treatment (**Fig 2B**). Origins that have lower levels of MCM in control cells are more prone to lose MCM signal in response to even intermediate levels of auxin treatment compared to origins that have high levels of MCM (**Fig 2B**). These results indicate that levels of MCM pools in cells

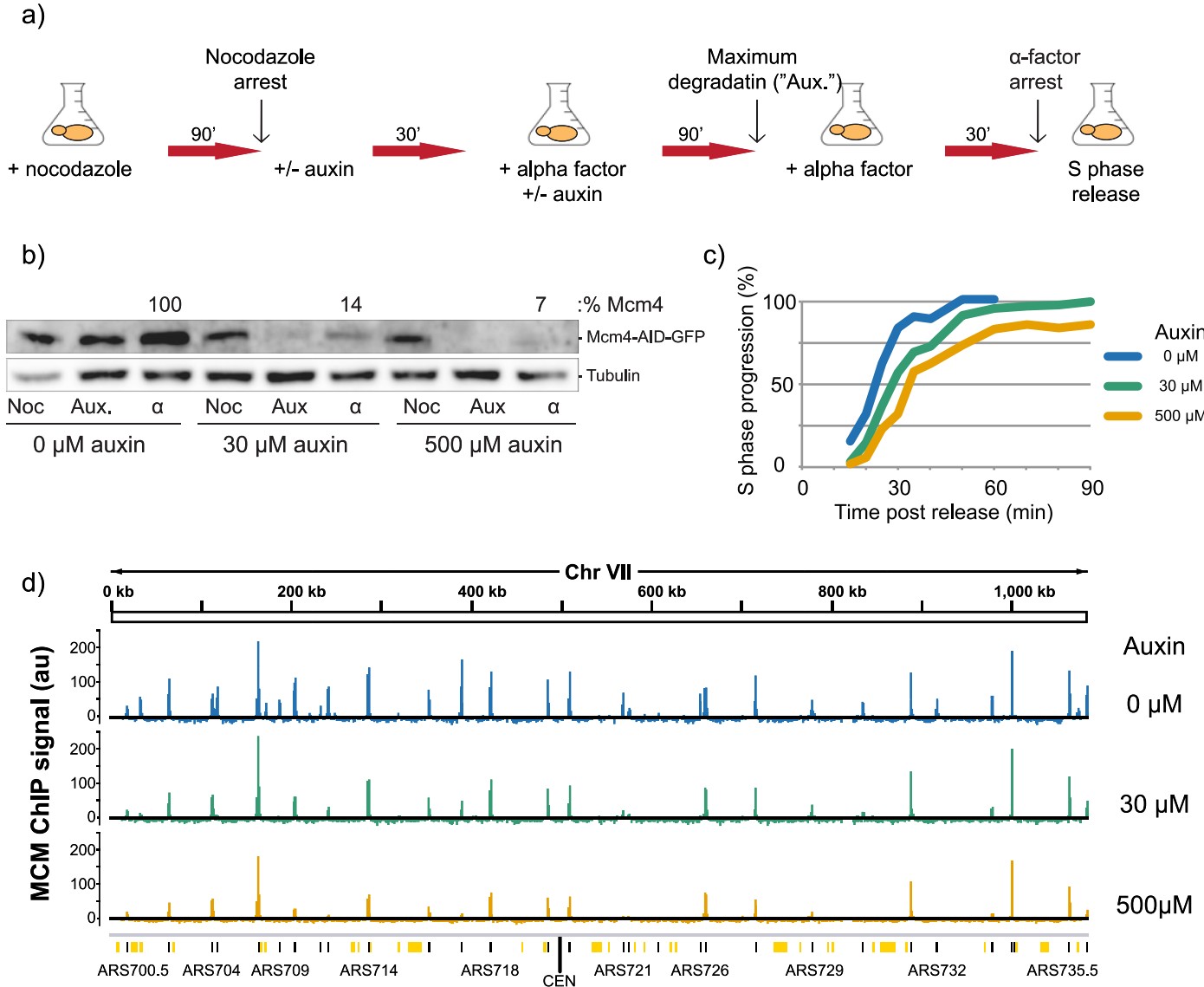

**Fig 1. Reduction of MCM pools through auxin-mediated degradation of Mcm4. a)** Experimental outline for MCM reduction experiments. yFS1059 cells were first synchronized with nocodazole, then released into α-factor for a G1/S arrest, prior to release into S phase. Flask symbol indicates filtering and release into new media, with additives as indicated. **b)** Western blot showing the levels of auxin-inducible cells (yFS1059) at the indicated time points of the experiment, probed using an anti-GFP antibody tracking GFP-tagged Mcm4. Noc = nocodazole arrest, Aux = maximum auxin degradation, α = α-factor arrest. Boxed value indicates quantitation of Mcm4-IAA17-GFP levels in α-factor arrested cells treated with 0 μM, 30 μM, or 500 μM auxin relative to levels of 0 μM auxin cells. **c)** Quantitation of flow cytometry data as cells progress from α-factor arrest through S phase. **d)** Normalized MNase-ChIP-seq coverage on Chromosome VII in 1 kb bins. ARS-annotated origins are shown as black lines whereas potential, non-experimentally-confirmed origins are shown as orange lines/boxes [63].

affect loading at origins of replication throughout the genome in significant and dissimilar ways.

To investigate the reason that reduced MCM levels delayed bulk S-phase progression, we examined we examined the distribution of MCM signal and inter-origin distances in the three conditions. We calculated the smallest number of origins required to account for 50% of the total MCM signal for each condition and measured the inter-origin distances between these

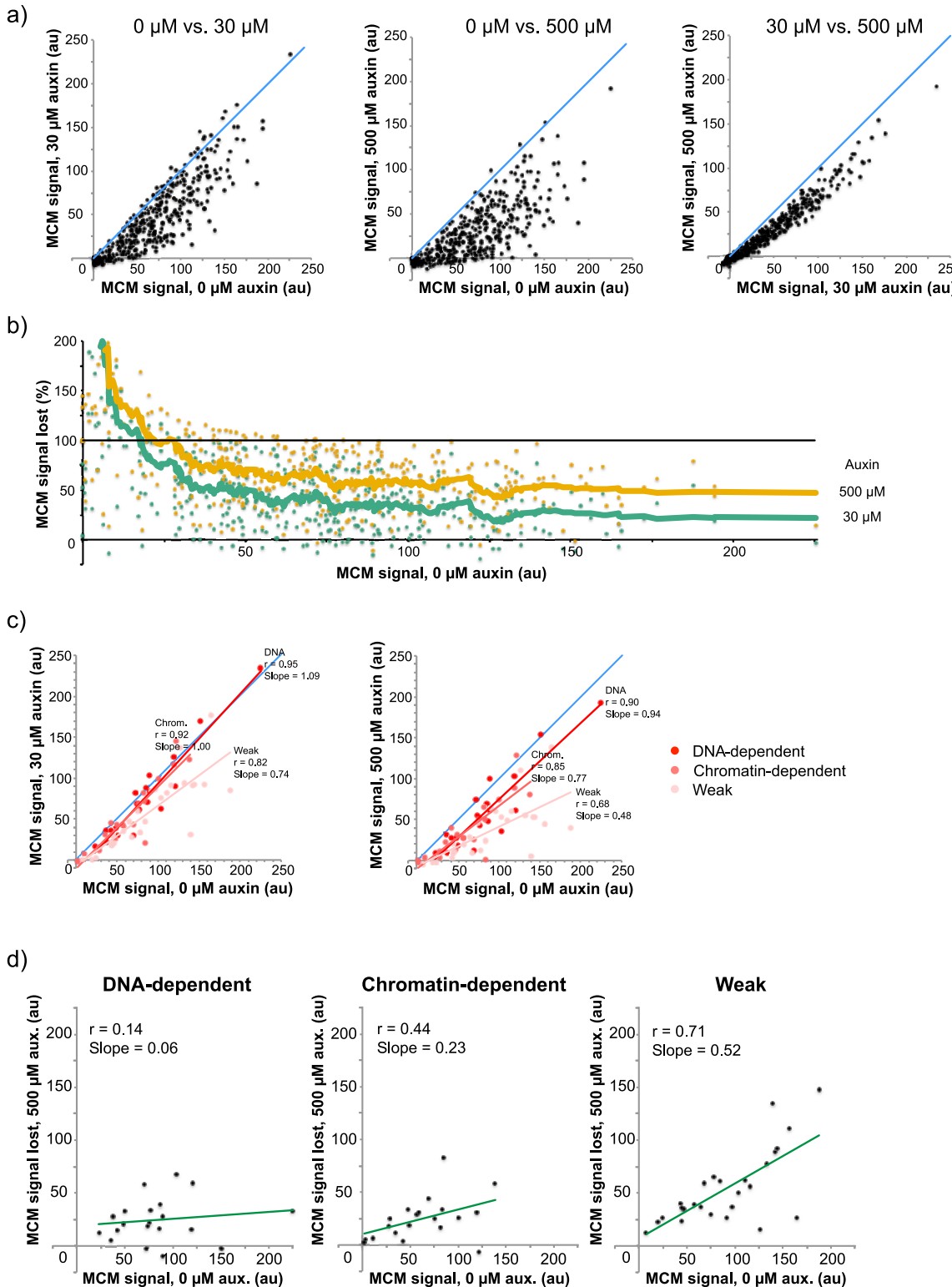

**Fig 2. Reduction of cellular MCM levels causes a reduction in helicase loading at origins of replication. a)** yFS1059 cultures were synchronized and released into S phase as shown in Fig 1A. MCM ChIP-seq signal in 1 kb bins around ARS-annotated origins from the indicated auxin treatments at the α-factor arrest point (x = y line drawn in blue for reference). **b)** Percent of MCM ChIP-seq signal lost upon treatment with 30 μM or 500 μM auxin. (20 pt moving average fit. y = 100 line drawn in blue for reference). **c)** MCM ChIP-seq signal at 0 μM auxin treatment compared to 30 μM and 500 μM auxin for origins of replication characterized as 'DNA-dependent',

'chromatin-dependent', and 'weak' in Hoggard et al. [31]. The regression lines are in the same colors as the data points they fit. x = y is shown in blue for comparison. **d)** MCM ChIP-seq signal for untreated cells plotted against the MCM ChIP-seq signal lost at the indicated auxin treatments for the origins indicated in **c**).

major origins. If MCM pools are reduced and loading is decreased significantly for some origins, others would be expected to replicate larger tracts of the genome, leading to an increase in the time required to complete S phase (**Fig 1C**). As seen in **S2 Table**, the smallest number of origins required to make up 50% of the MCM signal in each condition decreases as MCM pools are decreased (0 μM = 110, 30 μM = 85, 500 μM = 72), consistent with the observation that origins which load less MCM under control conditions lose a higher fraction of their MCM signal in response decreases in MCM pools. The *in-silico*-calculated average inter-origin distance between these top origins increases from 96 kb for 0 μM, to 118 kb for 30 μM, and to 136 kb for 500 μM auxin treatment, suggesting that they need to replicate larger tracts of the genome. In addition, a larger portion of origins replicate shorter tracts at 0 μM auxin, while a larger portion replicates longer tracts for 30 μM and 500 μM auxin treatments (**S4 Fig**). Overall, this data indicates that, in response to reduced MCM pools, origins that load higher levels of MCM are responsible for replicating larger tracts of DNA, leading to an increase in the amount of time required to complete DNA replication.

## 'Weak' origins are more prone to reduced MCM loading

Previous analysis of various replication origins in budding yeast classified origins based on their affinity for ORC *in vivo*, as measured by ChIP-seq, and *in vitro*, as measured by gel shift assays [31]. Origins for which high *in vitro* ORC affinity explained their high *in vivo* affinity were classified as DNA-dependent. Origins whose high *in vivo* affinity could not be explained by their *in vitro* affinity were classified as Chromatin-dependent. Origins that displayed low ORC affinity both *in vivo* and *in vitro* were classified as Weak. Importantly, the three classes of origins have similar distributions of efficiency (0.64±20, 0.69±19, 0.66±17, respectively) [32] and replication timing (21.8±2.1', 20.6±2.7', 21.4±2.1', this paper), showing that all three classes can support early/efficient and late/inefficient origins. To gauge the effect of lowered MCM pools on loading dynamics between these origin classes, we compared MCM levels for cells treated with 0 μM auxin to those treated with 30 μM or 500 μM auxin (**Fig 2C**). The comparison shows that DNA-dependent and Chromatin-dependent origins maintain most of their endogenous levels of loading in response to auxin treatment and therefore are more resistant to changes in MCM pools compared to Weak origins. As expected, Weak origins lose more of their signal in response to reductions in MCM pools compared to DNA- and Chromatin-dependent origins (**Fig 2D**). This analysis suggests that origins with low ORC affinity are outcompeted by origins with higher ORC affinity (due to sequence or chromatin differences) when the pool of available MCM is reduced.

## MCM associates with flanking nucleosomes and the nucleosome-free region (NFR) of origins

Studies using MNase footprinting indicate that MCM associates with well-positioned nucleosomes flanking the ACS at origins of replication [19]. To assess the location of MCM signal in our data, we constructed strand-specific V plots centered at the ARS consensus sequence (ACS) [11]. V plots map the abundance of reads and their length relative to their genomic location. Mapping MCM ChIP-seq reads from control cultures shows that MCM is mostly found within 2–3 nucleosomes from the ACS, with signal decreasing with distance away form the

ACS (**Fig 3A**). Examination of individual origins shows a similar distribution of MCMs (**S5A Fig**). The signal includes MCM-DH-sized reads (~68bp) as well as larger, nucleosome-sized reads (~146bp) (**Fig 3A**). Although most of the MCM ChIP-seq signal is associated with nucleosomes, as previously reported [19], the increased resolution of our approach also allowed us to detect MCM signal in the nucleosome-free region (NFR) of origins, as predicted by recent *in-vitro* cryo-EM structures [33]. In particular, the smaller, MCM-sized fragments, populate the NFR (**Figs 3A** and **S5B**). This observation was dependent on the degree of MNase digestion, as Replicate #1, which had a lower degree of digestion, lacked the smaller fragments, in agreement with similar studies involving transcription factors (**S6 Fig**)[34].

In order to gauge changes in MCM signal around origins in response to reduction in MCM levels we constructed read density profiles. To do so, we used the midpoint of ACS-annotated origins and quantified read densities based on their genomic location around these origins [11]. As expected, there was a reduction in MCM ChIP-seq read density at origins in response to both 30 μM and 500 μM auxin (**Fig 3B**). In contrast, there was a much smaller but reproducible increase in read density for the +1 and -1 nucleosome positions for input DNA, indicating a possible increase in nucleosome occupancy as MCM loading is reduced at these origins (**S7A Fig**). However, in a 1kb window around the origins there was no significant difference in read density for input samples (r = 0.96–0.98, **S7B Fig**). Given the differential response of origins in response to auxin, we separated origins in quartiles based on the percent of MCM signal that they lost in response to 500 μM auxin. As seen in **Fig 3C**, origins which are least affected by MCM pool reduction tend to have higher levels of MCM loaded under control conditions and do not lose any signal in response to intermediate levels of auxin. On the other hand, origins which are most affected by reduced MCM pools tend to have less MCM loaded under control conditions and upon intermediate auxin treatment lose most of their signal. This data elucidates sets of origins that are distinct in how they are affected by MCM levels, identifying origins that are resistant to intermediate reductions in MCM levels and others that are sensitive.

ORC binding to the ACS and subsequent MCM loading is a directional process dependent on an ACS-site and a similar but inverted nearby sequence [12,35]. To determine whether the directionality of MCM loading is maintained throughout G1 in relation to the dominant annotated ACS sites, we separated origins based on the magnitude of signal upstream or downstream of the ACS. The 253 origins tested were split nearly evenly between higher signal upstream or downstream of the annotated ACS (129 to 124, respectively; **S8 Fig**). The MCM signal was largely independent of ACS directionality, as a previously noted [19]. Only 50% of higher-upstream-signal origins and 44% of higher-downstream-signal origins correlated with ACS directionality. This data supports a model where, once loaded, MCM is able to passively slide in either direction on double stranded DNA, consistent with previous *in vitro* findings [33,36].

## Auxin-induced reduction in MCM loading causes significant delays in replication timing at corresponding origins

Reduction of MCM pools via auxin-induced Mcm4 degradation caused significant changes to MCM loading at origins. To test whether these changes in MCM abundance have an effect on replication timing, we measured replication timing in cells released into S phase after treatment with 0 μM, 30 μM, or 500 μM auxin, using the same cultures as those used for our ChIP-seq experiments. Briefly, using the sync-seq approach, we measured replication timing by measuring genome-wide copy number at multiple points after synchronous release into S phase [37]. From this data, we extracted the parameter $T_{rep}$, the time at which 50% of cells have replicated a specific locus. **Fig 4A** shows the $T_{rep}$ profile of Chromosome V along with the

a)

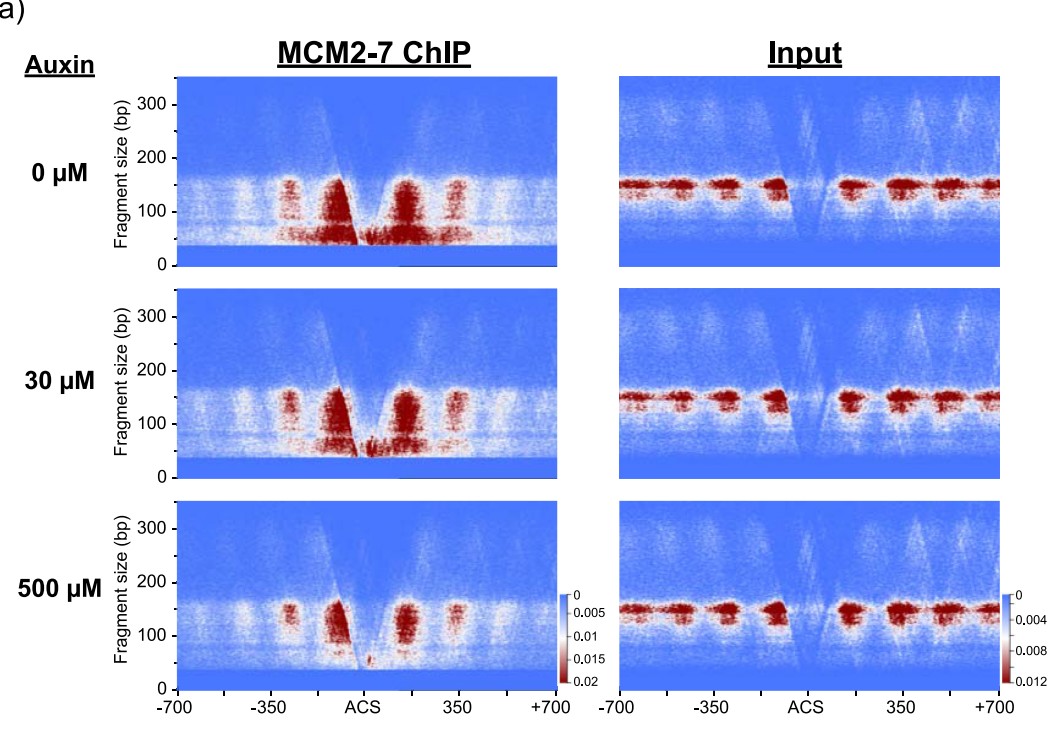

b)

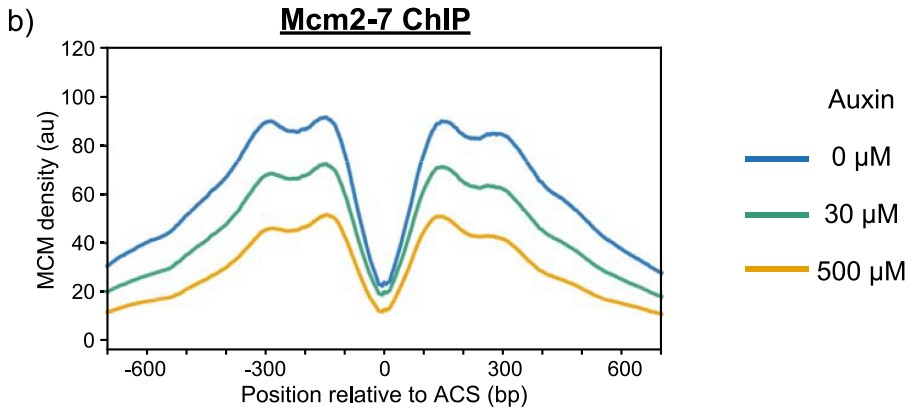

c)

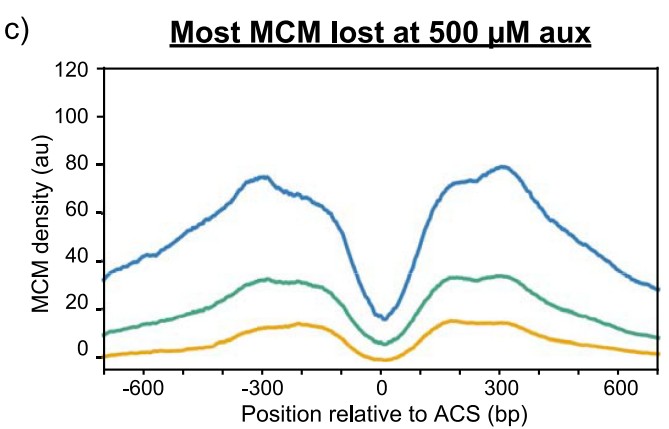 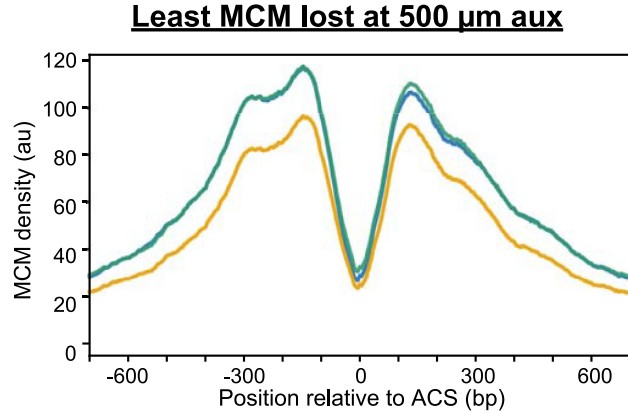

**Fig 3. MCMs associate with the origin-adjacent nucleosomes and the NFR and show differential loading dynamics at origins. a)** V plots of MCM MNase-ChIP-seq and input (non-IP) reads from yFS1059, plotted based on their length on a 700 bp window centered at 253 ACS-annotated origins [11]. **b)** Normalized MCM MNase-ChIP-seq density profiles centered at ACSs for the indicated auxin treatments. **c)** ACS-annotated origins were separated in quartiles based on the percent of signal lost in response to 500 μM auxin treatment compared to 0 μM auxin. The normalized coverage density profiles for the sets of origins displaying the largest reductions ('Most MCM lost at 500 μM auxin') and smallest reductions in MCM levels ('Least MCM lost at 500 μM auxin') are shown for all three treatments. Color scheme as in **b)**.

corresponding ChIP-seq profiles for treated and untreated cells. $T_{rep}$ values for cells expressing control levels of MCM correlate well with values obtained from previously published studies (r = 0.91, **S3C Fig**) [37]. A comparison of $T_{rep}$ values between untreated and auxin-treated conditions indicates that reduction of MCM levels leads to delays in replication timing across the genome and that these delays increase with higher auxin concentrations (**Fig 4B**). Together, this data demonstrates that reduction of cellular pools of MCM and subsequent reduction of MCM levels loaded at origins has direct implications for replication timing at specific origins in yeast.

Overlaying the $T_{rep}$ profiles reveals that some origins that are active in untreated cells become inactive in auxin-treated cells. Loss of origin activity from these origins correlates with a decrease in MCM signal by ChIP, as is evident for ARS512 and ARS520 (**Fig 4A**, black boxes). In contrast, other origins, such as ARS511 and ARS16, are less affected by auxin treatment (**Fig 4A**, gray boxes). To investigate the extent of this correlation, we examined it genome wide (**Fig 4C and 4D**). The replication timing of many origins throughout the budding yeast genome is controlled by specific mechanisms involving *trans*-acting factors and chromosomal location, such as Rif1, Rpd3, Fkh1, and Ctf19 binding, or telomere proximity, [16,38–41]. We measured $T_{rep}$ values and MCM abundance for the 46% of origins of replication that exclude those known to be affected by specific mechanisms of origin regulation [20]. Comparison of $T_{rep}$ values and MCM abundance for these origins under endogenous conditions indicated a small negative correlation between MCM abundance and timing of replication, suggesting that origins with higher MCM levels replicate earlier in S phase (**Fig 4C**, 0 μM auxin). The correlation increased and the slope became more negative in conditions where MCM levels were reduced via auxin (**Fig 4C**, 30 μM and 500 μM auxin). Furthermore, comparing changes in $T_{rep}$ with changes in MCM abundance as a result of auxin treatment revealed a positive correlation, indicating that larger decreases in MCM loading correlate with stronger delays in replication timing (**Fig 4D**). The analysis is complicated by the fact that $T_{rep}$ convolves origin firing time and passive replication. For instance, the MCM signal at ARS501 is greatly reduced and its replication-timing peak is lost (**Fig 4A**). However, its $T_{rep}$ value does not change much, because it is passively replicated by ARS511, which remains early-firing. Such passive replication reduces the effect seen in **Fig 4C and 4D**. Nonetheless, taken together, these results suggest that decreases in MCM loading lead to delays in replication timing at origins of replication. The 54% of origins that are known to be affected by *trans*-acting factors display a much weaker correlation between MCM signal and replication timing (r = 0.08, **S9A Fig**), but that correlation becomes stronger upon MCM depletion (r = 0.25–0.30, **S9B and S9C Fig**), suggesting that MCM loading affects the timing of these origins, as well.

## Cells overexpressing MCM are viable

Reduction of MCM pools leads to significant changes in MCM loading and replication timing. In order to investigate the effects of the converse condition of increased MCM pools, we employed the galactose overexpression system. The six genes encoding MCM were expressed from bidirectional Gal-1,10 promoters integrated into the genome, in addition to their endogenous copies [13]. The overexpressed copy of Mcm7 was tagged at its C terminus with GFP in

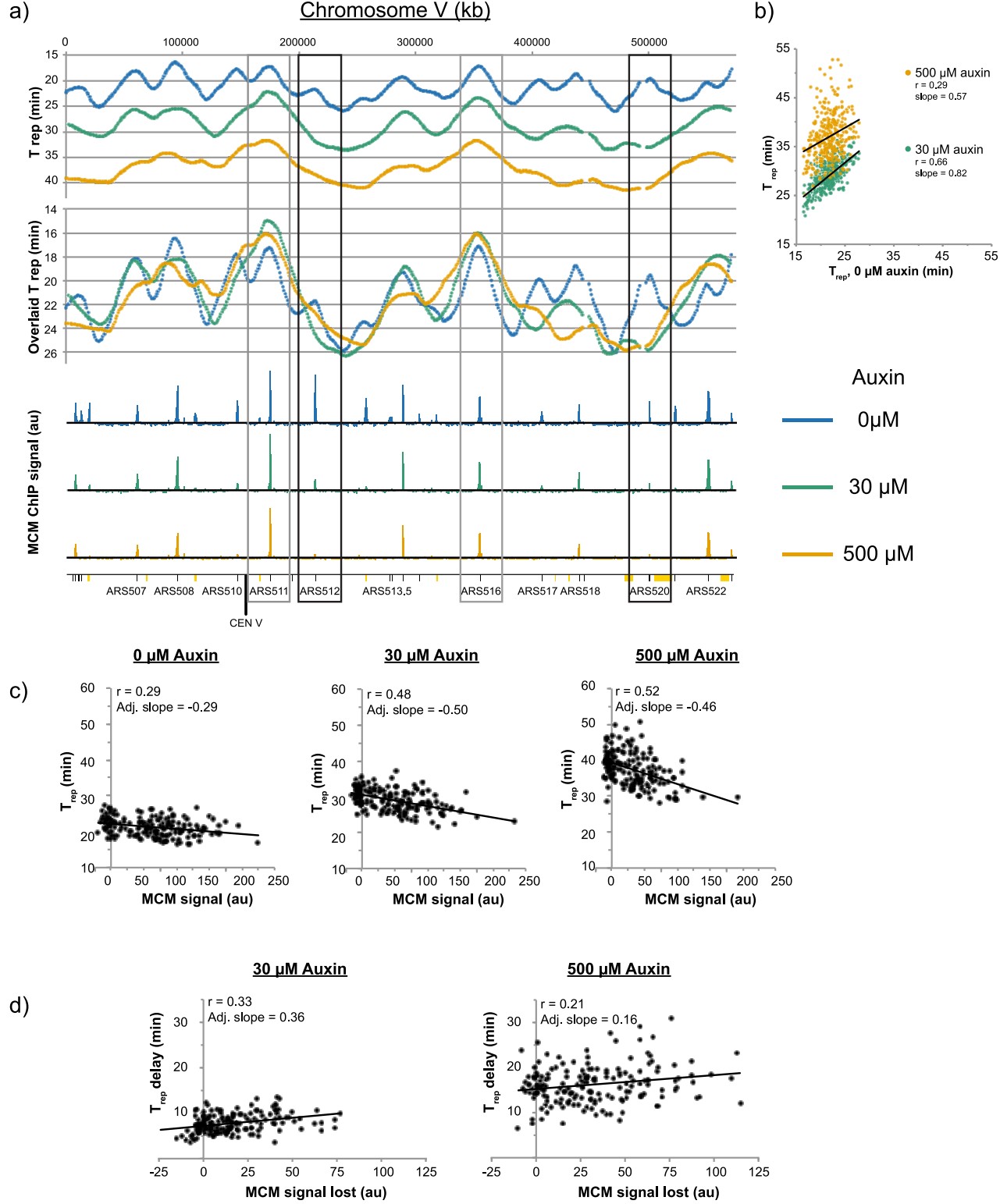

**Fig 4. Delays in replication timing correlate with reductions in MCM levels. a)** Chromosome V $T_{rep}$ values for the 0 μM, 30 μM, and 500 μM auxin-treated yFS1059 was calculated in 1kb windows, LOESS-smoothed, and plotted based on their genomic location. "Overlaid $T_{rep}$" values have been overlaid in order to compensate for delays in replication initiation and facilitate comparison between samples. Black boxes indicate two origins that display strong correlation between the loss of MCM signal and delay in replication timing. Gray boxes indicate two origins that display strong correlation between the maintenance of MCM signal and maintenance in replication timing. ARS-annotated origins are indicated in black dashes on

the bottom axis, whereas orange boxes denote unconfirmed origins from OriDB [63]. **b)** $T_{rep}$ values of ARS-annotated origins for untreated cells are plotted against values observed for 30 μM and 500 μM auxin treatments. **c)** Correlation between MCM signal and replication timing for origins that are not known to be affected by specific replication timing control factors [20]. **d)** Correlation between delay in replication timing as a result of auxin treatment and reduction in MCM signal, for origins plotted in **c**).

order to monitor its overexpression levels. To test whether cells overexpressing MCM are viable, we performed serial dilution assays on YP-Raffinose and YP-Galactose plates. Overexpression of the complete hexamer, with or without the added GFP tag on Mcm7, did not affect cell viability (**S10A Fig**). However, overexpression of the MCM hexamer in conjunction with the loading factor Cdt1, but not overexpression of Cdt1 alone, reduced viability. This data shows that Mcm2-7 overexpression by itself is not lethal and can be used to increase cellular pools of MCM helicase.

## Overexpression of MCM during a G1 arrest does not affect progression through S phase

We assayed whether overexpression of MCM leads to changes in helicase loading prior to S phase, and whether such changes affect replication timing. We synchronized cells in metaphase using nocodazole then released them into media containing α-factor for synchronization at the G1/S boundary (**Figs 5A** and **S1**). In addition to α-factor, the media was supplemented with galactose or raffinose, to induce Mcm2-7 overexpression or not, respectively. Induction of MCM was monitored via western blot using a polyclonal antibody against MCM. After two hours, a ~5-fold overexpression of the hexamer was observed (**Fig 5B**). A portion of the α-factor arrested culture was collected for ChIP-seq analysis while the rest was released into S phase to monitor replication timing. As seen in **Fig 5C**, cells overexpressing MCM during a G1 arrest show a slight delay in replication initiation in S phase. However, a control strain that did not overexpress MCM showed similar results, indicating that the delay was likely due to the switch in carbon source (**S10B Fig**). Therefore, overexpression of MCM does not appear to cause any significant changes to cell cycle progression.

## Increased MCM levels do not alter relative levels of helicase loading at origins

In order to measure MCM loading in cells overexpressing the full hexamer, we performed MNase ChIP-seq on α-factor arrested cells as outlined in **Fig 5A**. Fragments of the genome that were associated with MCM were pulled down, sequenced, and their abundance was normalized to an *S. pombe* spike-in control, resulting in peaks that correspond to previously annotated origins of replication (**Fig 5D**). In addition, the signal of MCM peaks correlates well with data from the MCM-depletion experiments, as well as previous publications (r = 0.78–0.93, **S11A** and **S11B Fig**).

To gauge whether increased cellular pools of MCM helicase lead to changes in loading during G1, we compared MCM abundance at origins of replication in cells overexpressing MCM and those that did not (**S1 Table**). MCM signal at origins in cultures overexpressing MCM show a ~41% decrease in abundance compared to cultures with endogenous MCM levels (**Fig 6A and 6E**). However, this difference is almost perfectly uniform throughout all origins assayed (r = 0.99, **Fig 6A**), and shows no apparent difference in MCM ChIP-seq reads (**Fig 6D**), pointing to technical reasons involving ChIP or non-MCM-specific biological consequences of the overexpression. This result was not due to an insufficient amount of time allowed for loading, as three hours of MCM overexpression in α-factor produced similar

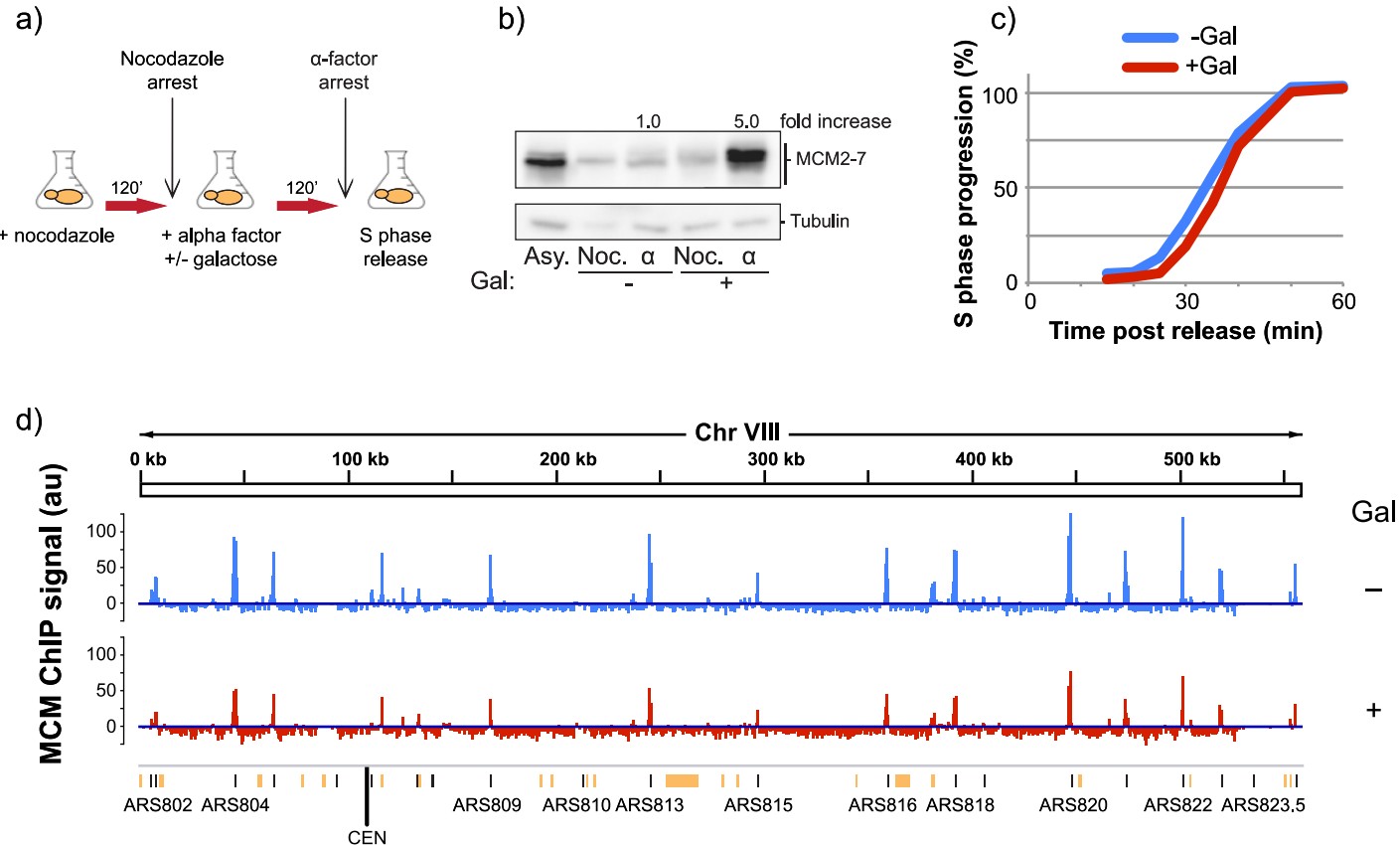

**Fig 5. Galactose-induced MCM overexpression. a)** Experimental outline for MCM overexpression experiments. yFS1075 cells were first synchronized with nocodazole, then released into α-factor for a G1/S arrest, prior to release into S phase. Flask symbol indicates filtering and release into new media, with or without galactose and additives as indicated. **b)** Western blot of yFS1075 cells showing the levels of MCM at the indicated time points of the experiment, probed using anti-Mcm2-7 polyclonal antibody UM174. Noc = nocodazole arrest, α = α factor arrest. Boxed values indicate quantitation of Mcm2-7 levels in α-factor arrested cells with or without galactose induction, relative to Mcm2-7 levels of uninduced cells. **c)** Quantitation of flow cytometry data of the replicating population of yFS1075 cells as they progress from α-factor arrest through S phase. **d)** Normalized MNase-ChIP-seq coverage on Chromosome VIII at the α factor arrest point in 1 kb bins. ARS-annotated origins of replication are shown as black lines whereas unconfirmed origins are shown as orange lines/boxes.

loading levels (r = 0.99, **S12A Fig**). As a result, we conclude that overexpression of Mcm2-7 does not alter relative helicase loading in G1.

MCM requires Cdt1 to enter the nucleus and to be loaded onto origins of replication [10]. Although Cdt1 disassociates from the MCM complex after it is loaded onto DNA, it doesn't fully exit the nucleus until late G1/early S phase. Therefore, although Cdt1 likely shuttles in and out of the nucleus in some equilibrium during the α-factor arrest in our experiments, it may not be able to accommodate nuclear import for all of the overexpressed MCM2-7 hexamers. To address the possibility that overexpressed MCM is not imported into the nucleus and therefore not loaded onto DNA due to substoichiometric levels of Cdt1, we overexpressed MCM2-7 along with Cdt1. However, we did not observe any significant changes in relative MCM loading (r = 0.98, **S12C Fig**). Furthermore, the lack of any changes was not due to the inability of the overexpressed hexamer to enter cell nuclei, as microscopy confirmed the presence of overexpressed Mcm7 in cell nuclei (**S13 Fig**). All together, these results indicate that overexpression of MCM helicase does not cause altered loading dynamics in G1.

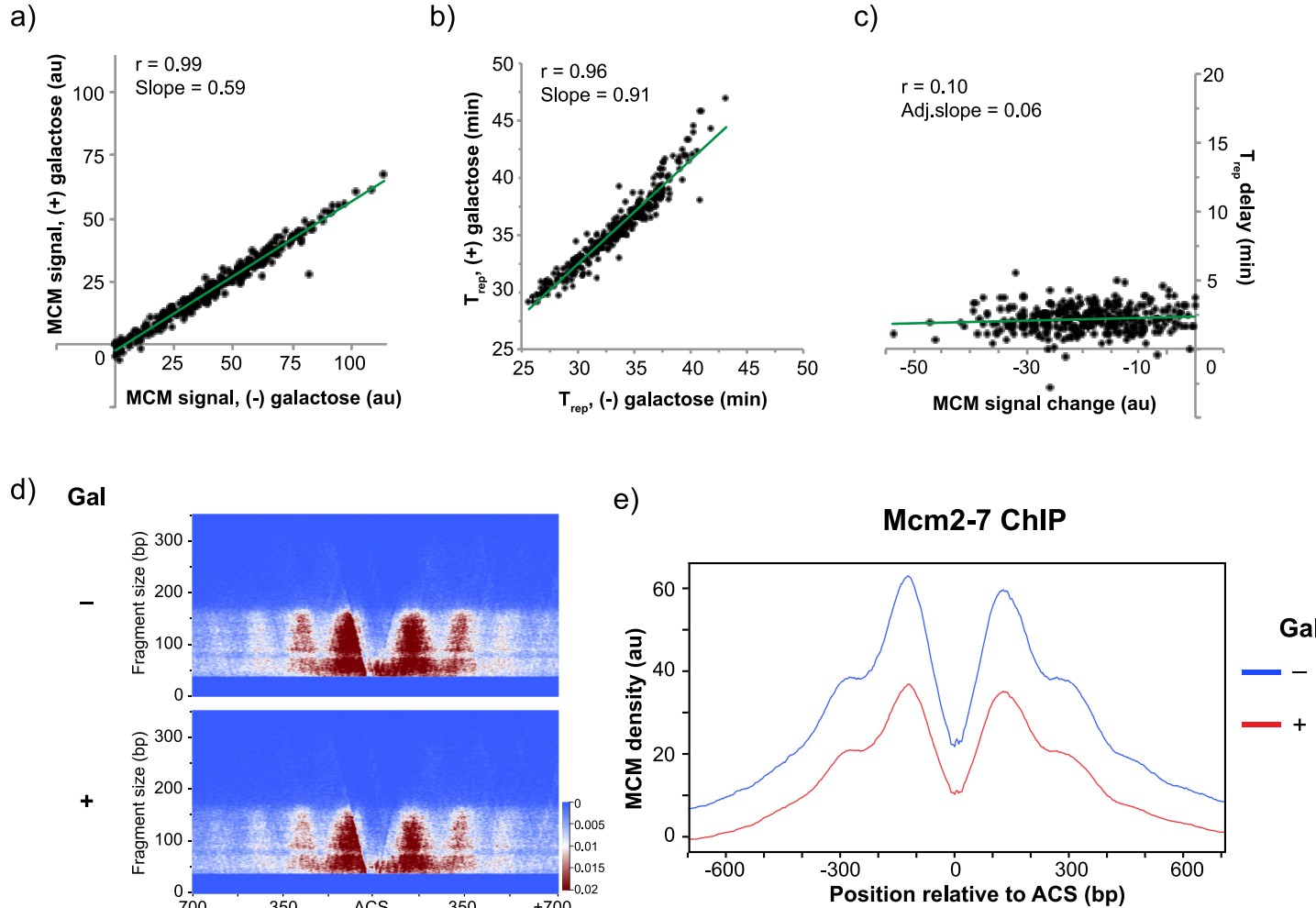

**Fig 6. MCM overexpression does not cause changes to helicase loading or replication timing. a**) yFS1075 cultures were synchronized and released into S phase as shown in Fig 5A. MNase-ChIP-seq signal at ARS-annotated origins at the α factor arrest point, with or without galactose-induced MCM overexpression. **b**) $T_{rep}$ values for cultures released into S phase as in Fig 5A, with or without prior galactose-induced MCM overexpression. **c**) Correlation between delays in replication timing and changes in MCM signal as a result of galactose-induced MCM overexpression. **d**) V plots of the MCM MNase-ChIP-seq reads, plotted based on their length on a 700 bp window centered at 253 ACS-annotated origins [11]. **e**) Normalized MCM MNase-ChIP-seq density profiles centered at ACSs with and without galactose-induced MCM overexpression.

### Overexpression of MCM does not affect replication timing

To test whether the increase in cellular MCM abundance led to changes in the replication timing program, we performed sync-seq experiments. $T_{rep}$ values for cells expressing endogenous levels of MCM correlate well with values obtained from the MCM-depletion experiments as well as previously published data (r = 0.92, **S11C Fig**) [37]. Moreover, origin replication times for cells overexpressing MCM do not significantly differ from control cells, beyond a brief and global delay attributed to the switch in carbon source (**Figs 6B** and **5C**). Moreover, changes in replication timing do not correlate with changes in MCM signal (**Fig 6C**). Altogether, this data shows that overexpression of the MCM helicase does not affect replication timing.

## Discussion

A unifying model of the mechanisms involved in establishing replication timing in eukaryotes remains elusive. Numerous *cis-* and *trans*-acting factors have been shown to affect timing, but

the mechanism by which they do so is unclear [8]. Since replication timing is primarily determined by the timing of origin firing, these factors are thought to affect the timing of MCM activation. However, it has also been noted that regulating MCM occupancy at origins would affect replication timing, because higher MCM occupancy at an origin would lead to a higher probability of it firing and thus an earlier average firing time [22,23,26]. Variable MCM occupancy could occur in one of two scenarios. The first is the single-MCM scenario, in which an origin can either be loaded with an MCM-DH complex or not. MCM occupancy at a specific origin would, in this scenario, be the fraction, from 0 to 1, of cells in which an MCM complex is loaded at that origin, a parameter that has been referred to as origin competency [21,24–26]. Alternatively, in the multiple-MCM scenario, more than one MCM-DH complex could be loaded at a single origin, so MCM occupancy could range from 0 to many [20,22,42].

Quantification of MCM components in budding yeast using a variety of techniques has yielded a wide range of values for the number of MCMs in each cell [43–45]. A meta-study using data from 21 publications reported median value estimates for MCM components that ranged from 2774–5360 molecules per cell for each of the components of the MCM helicase [46]. This analysis suggests that MCM is present in cells at levels that allow for loading of many more than one MCM at each of the 400 or so origins that are present in the genome. Moreover, measurements of *in vivo* chromatin binding in budding yeast indicate that MCM is loaded at a much higher number than accounted by the loading of a single MCM-DH at known origins [44].

Indeed, studies from various eukaryotic organisms have shown that MCM is loaded onto the genome at levels that are higher than necessary for replication under optimal conditions [47,48]. However, the excess of loaded MCM appears to be necessary under conditions that lead to elevated levels of replication stress. In *Xenopus* egg extracts, excess MCM loading becomes essential for successful replication under conditions of replication stress, indicating that dormant, excess MCM molecules are activated to salvage stalled replication forks that result from replication stress [49]. In *C. elegans*, worms only become susceptible to low doses of HU when their MCM levels are reduced [49]. Similarly, studies using human cells have suggested a mechanism where excess MCM hexamers that are loaded and not used during unperturbed replication become vital for successful replication after treatment with replication-stress-inducing agents [50,51]. Finally, MCMs were originally isolated based on the fact that reducing MCM number causes loss of minichromosomes, which are more sensitive to origin function than regular chromosomes [52,53]. Altogether, this data suggests that eukaryotes load more MCM onto DNA in G1 than is needed during an unperturbed S phase.

Although excess loading of MCM is important for a successful S phase under conditions of replication stress, it has also been implicated in contributing to the replication timing program under normal conditions. Genome-wide MCM mapping experiments in budding yeast have shown that MCM is loaded at origins in different amounts, and that the level of MCM loading correlates with the time in S phases at which an origin fires [20,21]. In addition, mutation of the B2 element in ARS1, which has been shown to be important for MCM loading at that origin, causes reduced MCM loading and a delay in replication timing [20,54]. This data points to a model where the more MCMs that are available for activation at an origin in S phase, the higher the likelihood that the specific origin will initiate early. Indeed, this MCM-stoichiometry model fits well with kinetic modeling of replication, which predicts that stochastic firing arising from differences in the availability of MCM in the presence of limiting initiation factors can give rise to the observed replication kinetics seen in budding yeast cells [21,22,26]. It is important to note that the MCM-stoichiometry models are equally valid whether MCM stoichiometry varies from 0 to 1 or 0 to many.

Our previous work supports a model in which multiple MCMs are loaded at origins [20,22]. Furthermore, overexpression of origin-activating factors in S phase causes most all origins to fire early in S phase, consistent with most origins having at least one MCM loaded [55]. Our MNase approach to MCM ChIP-seq reported here further supports the multiple-MCM model by producing high-resolution maps of MCM locations that identify MCMs at multiple locations across origins. We find MCM ChIP-seq signal distributed around origins up to three nucleosomes away from the ACS, similar to previous lower resolution results [19], in addition to signal inside the origin NFR (**Fig 3A**). The location of MCMs in the NFR and the surrounding nucleosomes support a model in which MCMs are mobile after being loaded and can slide past nucleosomes during nucleosome exchange. Indeed, this data is consistent with observations that early origins, which tend to have more MCM signal by ChIP-seq ([20] and Fig 4C), display higher rates of nucleosome exchange [56]. Furthermore, the fact that we find MCM with equal frequency upstream and downstream of origins suggests that ORC does not prevent MCM from diffusing past the ACS, consistent with ORC having a short dwell time at origins [13,57]. Nonetheless, because our current ChIP-seq data only represents an average of MCM locations, it cannot formally exclude the possibly that no more that one MCM-DH complex is ever loaded at a single origin—albeit at varying locations—and thus our current data does not discriminate between the single- and multiple-MCM scenarios. However, for all the analyses presented here, the two scenarios are formally equivalent and make the same predictions about how competition between origins for MCM loading affects replication timing.

Regardless of stoichiometry, the possibility of variable MCM occupancy raises interesting questions about how MCM occupancy is regulated and how it affects replication timing. We considered two hypotheses for how varying MCM occupancy between origins could be regulated. The first hypothesis, which we refer to as the ORC Activity model, posits that the rate at which ORC loads MCM varies among origins. ORC activity at an origin could vary due to ORC occupancy, which could be affected by ORC affinity for the ACS or by the local chromatin environment [31], or due to local regulation of ORC specific activity, which could be affected by local activators or repressors. The second hypothesis, which we refer to as the Origin Capacity model, posits that origins have an intrinsic capacity for MCM loading that varies among origins, and that once that capacity is achieved, no more MCMs can be loaded. To distinguish between these models, we varied cellular MCM levels, reasoning that changes in cellular MCM levels would lead to predictable changes in the level of MCM loaded at origins, depending on the mechanism that regulates origin loading. In particular, in response to a reduction in cellular MCM levels, the Origin Activity model predicts that all origins would be proportionally reduced in MCM occupancy, because origins will load at the same relative rates but the MCM pool will be depleted sooner. On the other hand, the Origin Capacity model predicts that origins with low capacities will fill up as normal, but that origins with higher capacities will fail to reach regular capacities if the MCM pool is depleted (**Fig 7A**).

Our results are not consistent with either the simple ORC Activity or Origin Capacity model (**Figs 2A** and **7A**). Instead, we propose a hybrid model, in which both ORC activity and origin capacity vary at each origin (**Fig 7B**). In the hybrid model, origin capacity is the primary effector of MCM occupancy in wild-type cells. ORC activity does vary among origins, but because neither time during G1 nor MCM amount is limiting, all origins are loaded to their full capacity. However, when MCM levels become limiting, not all origins can be fully occupied. In this case, origins with high ORC activity (bright red in **Fig 7B**) outcompete other origins and are still loaded to their full capacity, whereas origins with progressively lower ORC activity (light red and pink in **Fig 7B**) load MCM to progressively lower proportions of their full capacity before the MCM pool is exhausted. Therefore, although origin competition does

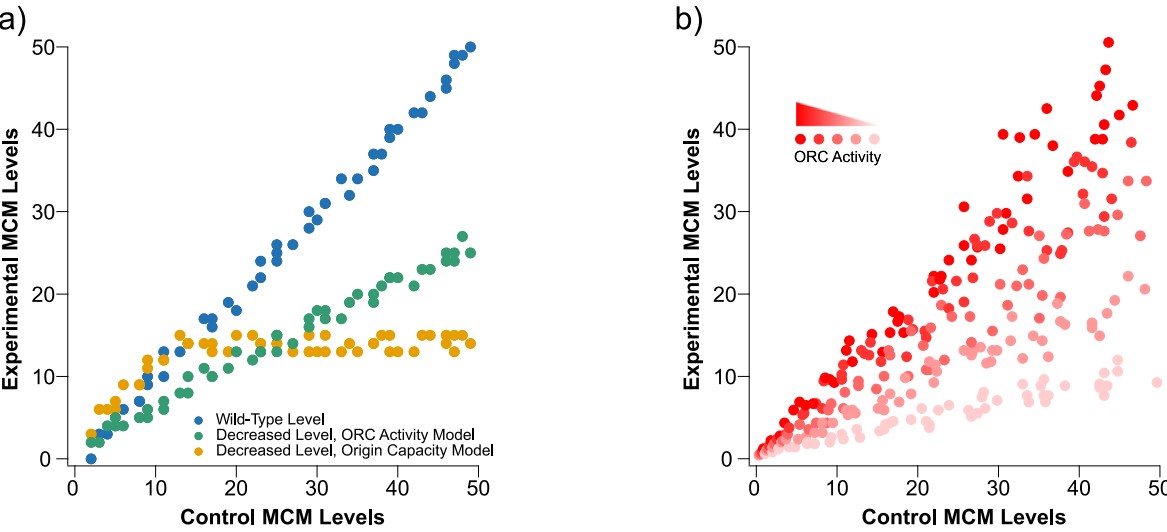

**Fig 7. Models for how changes in the cellular levels of MCM affect their loading dynamics at origins of replication and replication timing.**
**a)** Expected results if an ORC Activity model or an Origin Capacity model governs how MCM is distributed genome-wide. **b)** A combined model shows that, if there is a range of ORC activities and origin capacities, in normal cells ORC activity has little effect and MCM loading is largely determined by origin capacity because neither time nor MCM abundance are limiting. However, when MCM abundance becomes limiting, ORC activity becomes important because origins with more active ORC can load MCM faster.

not appear to play a decisive role in MCM loading in wild-type cells, its appearance when MCMs become limiting offers insight into the dynamics of MCM loading.

Our hybrid model is consistent with the behavior of three classes of origins previously characterized by ORC affinity and origin activity ([31] and Fig 2C). These three classes—DNA-dependent, chromatin-dependent and weak—show the behavior predicted by the hybrid model, with the strong, DNA-dependent origins maintaining close to full MCM loading (109–94%), the somewhat less strong, chromatin-dependent origins showing an intermediate reduction (100–77%), and the weak origins showing the greatest reduction in MCM loading (74–48%). Our hybrid model is also consistent with our MCM over-expression experiments. We find no change in the relative MCM ChIP-seq signal at origins in response to a five-fold increase in cellular MCM levels (r = 0.99, **Fig 6A**). This result is consistent with the conclusion that origins in wild-type cells are loaded to their full capacity and that MCM is normally in excess in budding yeast [44].

Although we find no relative difference in MCM ChIP-seq signal in our MCM overexpression experiments, we find an absolute reduction of 41% (**Fig 6A**). This result was dependent on overexpression constructs being present in the cells, because a control strain with no over-expressing genes did not display any reduction in MCM ChIP-seq signal in response to galactose (**S12B Fig**). This result does not appear to be due to a uniform increase in background signal, which might be expected if excess MCM was loaded with low specificity at many sites across the genome, because the data was normalized to an *S. pombe* spike-in to allow for absolute quantitation. It could be that the overexpression of MCM titrates away an interacting component that is crucial for MCM loading or nuclear import. However, in that case we would expect to see an effect similar to reducing MCM levels, which does not affect all origins uniformly (**Fig 2A**). Alternatively, although the antibody is added in saturating amounts, antibody-binding dynamics of chromatin-bound MCM could be altered due to the presence of excess MCM. In any case, given the lack of a plausible biological and mechanistic explanation

for reduced MCM loading in response to MCM overexpression, and the nearly perfect correlation between plus and minus overexpression conditions, we conclude that increased MCM levels do not cause altered MCM loading in yeast.

In addition to the lack of changes in MCM loading, overexpression of MCM did not affect cell viability, bulk S-phase progression, or replication timing (**Fig 6**). In contrast, overexpression of MCM along with Cdt1, but not Cdt1 alone, was lethal if constitutively expressed (**S8A Fig**), resulting in dumbbell-shaped terminal phenotype and suggesting a metaphase arrest. This result does not appear to be related to MCM loading or replication timing (at least in the window of one cell cycle) as levels of MCM loading and replication timing remained largely unchanged in MCM/Cdt1-overexpressing cells (**S11C Fig**). Although this points to a synergistic effect between MCM and Cdt1 in causing cell death, the exact mechanism for the lethality remains unclear.

Reduction of MCM levels caused a delay in progression through bulk S phase (**Figs 1C** and **4A**). The slower progression through S phase is consistent with less MCM being loaded onto some origins, resulting in fewer origins being activated in S phase. If MCM levels at all origins were uniformly reduced, we would not expect any change in replication timing because replication initiation in budding yeast is regulated by competition for a limited a set of initiation factors [15,58]. If MCM loading were lowered in uniform proportions, the relative competition for initiation factors, and thus replication timing, would be maintained. However, because the reduction of MCM loading is not uniform, and because low MCM origins tend to lose more MCM than high efficiency ones (**Fig 2B**), the distribution of MCM is more heterogeneous, leading to larger distances between efficient origins and thus longer S phase (**S4 Fig** and **S2 Table**).

Our combination of MCM ChIP-seq and replication timing data also allow us to directly confirm in a single experiment previous conclusions, drawn from comparing data collected in different studies, that replication timing correlates with levels of MCM loading [20,21]. Although the correlation between MCM levels and replication timing is robust (**Fig 4C**, $p > 10^{-5}$), it is clearly not the only determining factor (r = 0.29–0.52). Therefore, we conclude that MCM loading is a significant effector of replication timing, but one that is modulated by other factors that affect the specific activity of loaded MCMs. Nonetheless, by manipulating MCM levels and showing that replication changes concomitantly, we confirm genome-wide that MCM levels directly regulate replication timing (**Fig 4D**), an experiment that had previously been done only at a single locus [20].

In this study, we show that some origins are efficient at recruiting MCM, while others depend on excess levels of the helicase to display the loading observed in wild type cells. These results indicate that the relative levels of MCM loaded at origins are not simply static, but rather a dynamic balance dependent on cellular MCM levels, origin MCM recruiting ability, and origin MCM capacity. In addition, our data suggests that wild-type MCM abundance is saturating, such that increasing levels further does not lead to increased loading. Altogether, this work suggests that MCM loading homeostasis in wild type cells, and subsequently replication timing, is dependent on MCM being in excess. In future studies, it will be interesting to dissect the specific *cis-* and *trans*-acting factors that make origins sensitive or resistant to changes in MCM levels. The rapid evolution of origins within the *Saccharomyces sensu stricto* clade provides a rich resource for a genome-wide approach to such studies [59,60]

## Materials and methods

Strains used are listed in **Table 1**. Standard techniques were used for strain constructions. Cultures were grown in YP-Dextrose (YPD) at 30°C, unless otherwise noted.

**Table 1. Strains used.**

| Strain | Genotype | Source |
|---|---|---|
| yFS105 | h- leu1-32 ura4-D18 | Lab WT *pombe* strain |
| yFS833 | α leu2-3,112 ura3 trp1-1 ade2-1 can1-100 his3-11,15 GAL+ ssd1-d2 RAD5+ | [37] |
| yFS1020 | a leu2-3,112 ura3-1 trp1-1 ade2-1 can1-100 his3-11,15 bar1-Δ lys2::HisG Pep4Δ | [13] |
| yFS1021 | a leu2-3,112 ura3-1 trp1-1 ade2-1 can1-100 his3-11,15 bar1::HisG lys2::HisG Pep4Δ Trp1::Gal1,10-MCM6,MCM7 His3::Gal1,10-MCM2,MCM3 Lys2::Gal1,10-MCM4, MCM5 Ura3::Gal1,10-Cdt1,Gal4 | [13] |
| yFS1044 | a leu2-3,112 ura3-1::ADH1-osTIR1-2-9myc (URA3) trp1-1 ade2-1 can1-100 his3-11,15 | [27] |
| yFS1059 | a leu2-3,112 ura3-1::ADH1-osTIR1-2-9myc (URA3) trp1-1 ade2-1 can1-100 his3-11,15 MCM4::MCM4-IAA17-GFP(HPH) | This study |
| yFS1062 | a his3-11,15 leu2-3,112 ura3-1::ADH1-osTIR1-2-9Myc(URA3) trp1-1 ade2-1 can1-100 Mcm4::Mcm4-IAA17(kanMX) | This study |
| yFS1075 | a leu2-3,112 ura3-1 trp1-1 ade2-1 can1-100 his3-11,15 bar1-Δ lys2::HisG Pep4Δ Trp1::Gal1,10-MCM6,MCM7-GFP His3::Gal1,10-MCM2,MCM3 Lys2::Gal1,10-MCM4, MCM5 | This study |
| yFS1076 | a leu2-3,112 ura3-1 trp1-1 ade2-1 can1-100 his3-11,15 bar1-Δ lys2::HisG Pep4Δ Trp1::Gal1,10-MCM6,MCM7 His3::Gal1,10-MCM2,MCM3 Lys2::Gal1,10-MCM4,MCM5 | This study |
| yFS1080 | a leu2-3,112 ura3-1 trp1-1 ade2-1 can1-100 his3-11,15 bar1-Δ lys2::HisG Pep4Δ Ura3::Gal1,10-Cdt1,Gal4 | This study |
| yFS1081 | a leu2-3,112 ura3-1 trp1-1 ade2-1 can1-100 his3-11,15 bar1-Δ lys2::HisG Pep4Δ Ura3::Gal1,10-Cdt1,Gal4 Trp1::Gal1,10-MCM6,MCM7 His3::Gal1,10-MCM2,MCM3 Lys2::Gal1,10-MCM4-GFP,MCM5 Ura3::Gal1,10-Cdt1,Gal4 | This study |
| yFS1082 | a leu2Δ0 ura3Δ0 his3Δ1 met15Δ0 MCM4-GFP(his3mx6) | [65] |
| yFS1083 | a leu2Δ0 ura3Δ0 his3Δ1 met15Δ0 MCM7-GFP(his3mx6 | [65] |

## Auxin degradation experiments

*S. cerevisiae* strain yFS1059 (MCM4-AID-GFP) was grown to an OD of ~0.2. Nocodazole was added to a final concentration of 7.5 μg/ml for two hours. 1.5 hours into the nocodazole arrest, cultures were supplemented with auxin (Indole-3-acetic acid sodium salt, Sigma I5948) to final concentrations of 500 μM, 30 μM, or 0 μM. After two hours total in nocodazole, cultures were vacuum-filtered and cells were resuspended in fresh YPD supplemented with 5 μg/ml α-factor (Biomatik) and auxin to a final concentration of 500 μM, 30 μM, or 0 μM, as before filtration. After 1.5 hours, cultures were vacuum filtered and cells were resuspended in fresh YPD supplemented with 500 μM, 30 μM, or 0 μM, as before filtration, for 30 minutes. Cultures were vacuum-filtered a final time and cells were resuspended in fresh YPD for S phase progression.

For ChIP sample collection, α-factor-arrested cells were crosslinked with formaldehyde and collected as previously described (Wal & Pugh 2012).

## Galactose overexpression experiments

*S. cerevisiae* strains yFS1075 (GAL-MCM), yFS1021 (GAL-MCM, GAL-CDT1), and yFS1020 (WT) were grown in YP-Raffinose (YP-Raf) at 30˚C to an OD of ~0.2. Nocodazole was added to a final concentration of 10 μg/ml. After two hours, cultures were vacuum-filtered and cells were resuspended in fresh YP-Raf supplemented with α-factor (Biomatik) to a final concentration of 25 nM, as well as either galactose to a final concentration of 2% or an equal volume of water. Cells were allowed to arrest in α-factor for either two or three hours, followed by release via vacuum filtration into fresh YPD supplemented with 0.2 mg/ml pronase (Sigma P6911).

For ChIP sample collection, α-factor arrested cells were crosslinked with formaldehyde and collected as previously described (Wal & Pugh 2012). After formaldehyde quenching, *S. cerevisiae* cells were combined with log-phase wild-type *S. pombe* cells (yFS105, also crosslinked) at a 9:1 ratio.

## Fluorescence microscopy

Cultures were grown and supplemented with 2% final galactose as in **Fig 5A**. At the α-factor-arrest point, 1ml cells were fixed by the addition of 10ml methanol which was pre-chilled in dry ice, then stored at -80˚C until imaging. To prepare for imaging, fixed cells were equilibrated to 4˚C from their -80˚C storage, centrifuged, and all but 1ml supernatant was removed. The remaining mix was washed 2x with 2 ml ice cold PBS. Cells were then resuspended in 150 μl Vectashield+DAPI (Vector Laboratories H-1200) and imaged using a Zeiss Axioskop 2 Plus microscope fitted with DIC, GFP, and DAPI filters.

## Spot assays

Yeast was grown in YPD (auxin strains) or YP-Raf (Galactose-overexpression strains) overnight at 30˚C. Log phase cells were collected by brief centrifugation, washed twice with PBS, serially diluted five-fold starting with $4x10^6$ cells, then spotted onto the specified medium. Plates were incubated at 30˚C for at least 36 hours before imaging.

## Western blots and quantitations

Standard western blot techniques were used [61]. Mcm2-7 was probed with polyclonal UM174 (gift from Bell lab, MIT—[62]) and anti-rabbit HRP conjugate secondary (Promega W401B), tubulin was probed with monoclonal anti-tubulin antibody (Sigma T5168) and anti-mouse HRP conjugate secondary (Promega W402B), and GFP was probed with the monoclonal JL8 antibody (Takara) and anti-mouse HRP conjugate secondary (Promega W401B). Probing of Mcm2-7 using UM174 antibody was performed using 12% gels in order to compress signal from the six MCMs into a compact region to facilitate quantification. Signal was detected using SuperSignal West Dura chemiluminescent HRP substrate (Thermo 34076) and GE Amersham Imager 600. Quantitation of gel images was carried out using ImageJ (NIH, Maryland, USA).

## ChIP experiments

ChIP experiments were carried out as described previously with modifications noted below (Wal & Pugh 2012). MNase was titrated for each sample to determine concentrations that would yield ~80–90% mononucleosomal digestion. Immunoprecipitation was carried out using 95% of the MNase digested sample (v/v) and 4 μl of anti-MCM2-7 polyclonal antibody (UM174, gift from Bell lab, MIT—[62]). The remaining 5% of MNase digested sample was processed as 'input' and was treated with Proteinase K and RNase A to prepare libraries for deep sequencing.

## Flow cytometry

0.25 ODs of fixed cells (see below) were washed once with water, resuspended in 250 μl RNase A solution (100 μg/ml RNase A, 50 mM Tris pH 8.0, 15 mM NaCl), and incubated overnight at 37˚C. Cells were then collected and resuspended in Proteinase K solution (125 μg/ml Proteinase K, 50 mM Tris pH 8.0) for 1 hour at 50˚C. The samples were then collected by centrifugation, resuspended in 1 ml staining solution (1 μM Sytox green–ThermoFisher S7020, 50

mM Tris pH8.0), sonicated briefly using a microtip sonicator, and analyzed for flow cytometry using a Guava easyCyte instrument.

### Sync-seq replication timing experiments

Experiments to monitor replication timing consisted of measuring population movement through S phase by flow cytometry and measuring genome-wide replication timing by deep sequencing [37]. To do so, 3 ODs of cells were collected for each time point and arrested by the addition of sodium azide to 0.1% and EDTA to 20mM. From these samples, 0.25 ODs were used for flow cytometry and the rest was used for genome-wide copy number analysis by deep sequencing.

To prepare genomic DNA for deep sequencing, cells were lysed as for ChIP experiments. Lysates were treated with proteinase K and RNase A, followed by two consecutive extractions with phenol-chloroform-IAA (25:24:1, Fisher). DNA was purified by ethanol precipitation and resuspended in 135 μl water in preparation for shearing. A Covaris machine was used to shear the DNA to an average size of ~200bp, following manufacturer's instructions. DNA was then purified by DNA Clean and Concentrator Columns (Zymo Research) before making libraries.

### Library preparation and sequencing

Libraries were prepared using NEBNext Ultra II kits (NEB) following the manufacturer's protocol. Following library preparation, ChIP samples were purified as described [30], while input samples were purified using a 2:1 ratio of AmpureXP (GE) beads to library. Following PCR amplification, all samples were purified using AMpure XP beads with 0.9:1 beads to PCR ratio. Samples were sequenced using the Illumina NextSeq 500 platform with paired ends. See S3 and S4 Tables for library statistics.

### Data analysis

The sequencing data from this study have been submitted to the NCBI Sequence Read Archive under accession number PRJNA663099. For MNase-ChIP experiments, sequencing reads were mapped to SacCer3 using Bowtie1, a 650bp upper limit cutoff, and exclusion of reads mapping to more than one genomic location. Duplicates were removed using Samtools 1.4.1. ACS-centered density profiles were constructed using coverage files normalized as indicated in the paragraph below. V plot intensities were normalized to counts per million (CPM) and generated by taking into account strand specificity of the ACSs [11] using R and modified in-house scripts (courtesy of Nils Krietenstein). Heatmaps were generated using Deeptools 3.0.2 without taking strand specificity into account. Quantification of MCM signal at origins of replication was done using a 1 kb window centered at the origin or ACS midpoints. Correlation coefficients and slopes were calculated for origins displaying normalized MCM signal greater than zero for untreated cells. The set of origins displaying ARS activity in plasmid assays from OriDB [63] was used for all quantitations, unless otherwise stated. For the MCM depletion experiments, all analyses were preformed on both replicates with similar results. Results from Replicate #1 are presented for all analyses because it displayed a better separation between the 30 and 500 μm datasets. The exception is the V plots, for which the higher resolution Replicate #2 is presented.

Coverage files for auxin degradation experiments were normalized to CPM and to the non-origin background signal in 25 bp windows. Input signal was then subtracted from the normalized ChIP coverage files to account for MNase digest-induced background oscillations. For galactose induction experiments, ChIP coverage files were normalized in 25 bp windows to the total number of reads mapped to the *S. pombe* spike-in control. Normalized input signal was

then subtracted from the normalized ChIP coverage to account for background oscillations in the data.

For replication timing experiments, coverage files were generated using LocalMapper scripts and $T_{rep}$ was calculated in 1000 bp bins genome wide using Repliscope. Both packages were shared by Dzmitry Batrakou of the Nieduszynski Lab and are available at https://github.com/DzmitryGB/ [64]. $T_{rep}$ data generated from Repliscope was LOESS-smoothed in windows of ~50kb using Igor (Wavemetrics, Lake Oswego, OR, USA). $T_{rep}$ data for subtelomeric origins residing within 15 kb of chromosome ends was discarded due to large amount of noise. At least seven samples covering S phase progression were used to estimate $T_{rep}$.

## Supporting information

**S1 Fig. Progression of synchronized cultures through S phase as monitored by flow cytometry.** yFS1059 cultures (replicates 1 and 2) were treated and synchronized as shown in **Fig 1A** before being released into S phase for replication timing measurements. yFS1075, yFS1020, and yFS1021 cultures were treated and synchronized as indicated in **Fig 5A** before being released into S phase for replication timing measurements. For yFS1075, flow cytometry data is shown for experiments where galactose-induced overexpression was performed for two hours or three hours. For yFS1020 and yFS1021, overexpression was performed for two hours. (PDF)

**S2 Fig. Auxin-induced degradation of Mcm4 causes reduced viability. a)** For chronic down-regulation of MCM, exponentially-growing cultures of the indicated genotypes were serially diluted and spotted on plates containing 0 μM, 30 μM, or 500 μM auxin, with or without 100 mM hydroxyurea. OsTIR (yFS1044) refers to the background of the strain required for auxin-mediated degradation. WT = yFS833; MCM4-IAA17 = yFS1062; MCM4-IAA17-GFP = yFS1059; MCM4-GFP = yFS1082. **b)** For acute down-regulation of MCM, yFS1059 cultures were arrested at G1/S using α-factor as outlined in **Fig 1A**, washed, then serially diluted on YPD plates with or without 100 mM HU. (PDF)

**S3 Fig. MNase-ChIP-seq and replication timing results from the MCM reduction experiments are reproducible. a)** MCM ChIP-seq signal at ARS origins in two biological replicates of yFS1059 for 0 μM, 30 μM, and 500 μM auxin treatments. **b)** Comparison of MCM signal at origins between the 0 μM auxin condition in this study and other publications [18,42]. **c)** Genome-wide replication timing correlation in 1 kb windows between the 0 μM auxin condition in this study and [59]. (PDF)

**S4 Fig. Genome-wide reduction in MCM loading leads to larger inter-origin distances. a)** The smallest number of origins required to make up 50% of the total MCM signal was calculated for each condition. The inter-origin distances for these sets of origins were calculated and plotted as a fraction of the total number of inter-origin distances for each condition. (PDF)

**S5 Fig. Effects of MCM depletion on MCM loading at origins.** a) MCM signal at well-known and example single origins in response to auxin treatment (yFS1059—Replicate 2). Signal at the well-known ARS1 and ARS501 origins of replication shows slight reductions in response to treatment with 0 μM, 30 μM, and 500 μM auxin. ARS512 loses most of its MCM signal in response to auxin treatment, while ARS516 signal remains largely the same. b) Normalized MCM MNase-ChIP-seq density pro les centered at ACSs for the indicated auxin treatments

(yFS1059—Replicate 2) for all, MCM-sized (50–90 bp) or nucelosome-sized fragments (125–165 bp).
(PDF)

**S6 Fig. Fragment analyzer results for the two MCM reduction replicates shows different degrees of MNase digestion.** Fragment Analyzer (Advanced Analytical Technologies) results of sequencing-ready input libraries show the different degrees of digestion for Replicates #1 and #2 of yFS1059. Replicate #2 displays higher mononucleosome-sized (146bp nuc-fragment + sequencing adapters and primers = ~284bp) content and therefore a stronger digestion.
(PDF)

**S7 Fig. Input density profiles and origin coverage quantitations do not change significantly after auxin-induced Mcm4 degradation. a)** Input (non-IP) read coverage density profiles at ARS origins of replication for Replicates #1 and #2 of yFS1059 for the indicated auxin treatments. **b)** Comparison of input coverage within 1 kb of ARS origins between 0 μM versus 30 μM and 500 μM auxin treatments.
(PDF)

**S8 Fig. ACS-annotated origins separated by magnitude of signal upstream or downstream of ACS.** MCM ChIP-seq signal in yFS1059 was quantified 500 bp upstream and downstream of ACSs. Origins were separated based on the dominant signal. Heatplots were generated for the set of origins displaying higher MCM signal upstream of the ACS as well as for the set of origins displaying higher signal downstream of the ACS.
(PDF)

**S9 Fig. Correlation of MCM signal and replication timing for origins that are affected by specific timing mechanisms.** Correlation between MCM signal and replication timing for origins that are known to be affected by specific replication timing control factors [20].
(PDF)

**S10 Fig. MCM overexpression does not affect cell viability. a)** Serial dilution assay of exponentially growing cultures of the indicated overexpressing genotypes, grown in the presence of raffinose or galactose. (-) = yFS1020, MCM2,3,4,5,6,7(GFP) = yFS1075; MCM2,3,4,5,6,7, CDT1 = yFS1021; CDT1 = yFS1080; MCM2,3,4,5,6,7 = yFS1076. **b)** Cultures of yFS1020, a control strain with no overexpression vectors, were treated as in **Fig 5A** and released into S phase. Flow cytometry quantitation shows the S phase progression of cells supplemented with glucose during the α-factor arrest versus those there were not.
(PDF)

**S11 Fig. MNase-ChIP-seq and replication timing results from the MCM overexpression experiments are reproducible. a)** MCM ChIP-seq signal at ARS origins for the two main 'untreated' samples in this study: 0 μM auxin condition (**Fig 1A**–yFS1059) versus (-) galactose condition (**Fig 5A**–yFS1075). **b)** Comparison of MCM signal at origins between the (-) galactose condition in this study (yFS1075) and other publications [18,42]. **c)** Replication timing correlation between the (-) galactose condition (yFS1075) and: i) 0 μM auxin condition for ARS origins, and ii) genome-wide 1 kb windows from Mueller et al [59].
(PDF)

**S12 Fig. Overexpression of MCM for longer times or along with Cdt1 does not alter loading levels at origins or replication timing. a)** Quantitation of MCM signal at ARS origins when MCM is overexpressed as in **Fig 5A** but for 3 hours (yFS1075). **b)** Quantitation of MCM signal at ARS origins for a control strain that does not overexpress any genes (yFS1020). **c)**

Quantitation of MCM signal and $T_{rep}$ at ARS origins when MCM is overexpressed along with Cdt1 as in **Fig 5A** (yFS1021).
(PDF)

**S13 Fig. Overexpressed MCM is able to enter the nucleus during α-factor arrest.** WT (yFS1020), MCM7-GFP (yFS1083), MCM2, 3, 4, 5, 6, 7-GFP overexpressing (yFS1075), and MCM2, 3, 4-GFP, 5, 6, 7 and CDT1 overexpressing (yFS10810) cultures were grown as in **Fig 5A** in the presence of galactose and visualized at the α factor arrest time point using DAPI to stain DNA and GFP fluorescence to visualize MCM quantity and localization.
(PDF)

**S1 Table. Quantification of MCM signal at origins of replication and Trep data for ARS-annotated origins (OriDB).**
(XLS)

**S2 Table. Smallest number of origins that make up 50% of total MCM signal for each condition, as well as the interorigin distances between those sets of origins.**
(XLS)

**S3 Table. Mnase-ChIP Sequencing Statistics.**
(XLS)

**S4 Table. Replication Timing Sequencing Statistics.**
(XLS)

**S5 Table. Numerical data for all figures.**
(XLS)

## Acknowledgments

We thank Nils Krietenstein for his help with sequencing and data analysis and Dzmitry Batrakou for his help with his Repliscope package. We also thank Steve Bell for sharing his MCM anti-sera.

## Author Contributions

**Conceptualization:** Livio Dukaj, Nicholas Rhind.

**Data curation:** Livio Dukaj.

**Formal analysis:** Livio Dukaj, Nicholas Rhind.

**Funding acquisition:** Nicholas Rhind.

**Investigation:** Livio Dukaj.

**Methodology:** Livio Dukaj.

**Project administration:** Nicholas Rhind.

**Software:** Livio Dukaj.

**Supervision:** Nicholas Rhind.

**Validation:** Livio Dukaj, Nicholas Rhind.

**Visualization:** Livio Dukaj, Nicholas Rhind.

**Writing – original draft:** Livio Dukaj, Nicholas Rhind.

**Writing – review & editing:** Livio Dukaj, Nicholas Rhind.

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
