## [Editor Report · Decision Letter 0]

14 Dec 2020

Dear Dr Rhind,

Thank you very much for submitting your Research Article entitled 'Competition for MCM Loading at Origins Establishes Replication Timing Patterns' to PLOS Genetics.

The manuscript was evaluated at the editorial level. At this stage, the editors would like to invite you to address the remaining comments from the Review Commons reviewers, and to submit a fully revised manuscript. The revised manuscript will then be editorially evaluated, and if necessary the editors will seek comments from one or more of the original reviewers.

If you decide to revise the manuscript for further consideration at PLOS Genetics, please aim to resubmit within the next 60 days, unless it will take extra time to address the concerns of the reviewers, in which case we would appreciate an expected resubmission date by email to plosgenetics@plos.org.

[LINK]

We are sorry that we cannot be more positive about your manuscript at this stage. Please do not hesitate to contact us if you have any concerns or questions.

Yours sincerely,

Gregory P. Copenhaver

Editor-in-Chief

PLOS Genetics

Gregory S. Barsh

Editor-in-Chief

PLOS Genetics

---

## [Editor Report · Decision Letter 1]

4 Mar 2021

Dear Dr Rhind,

We are pleased to inform you that your manuscript entitled "The Capacity of Origins to Load MCM Establishes Replication Timing Patterns" has been editorially accepted for publication in PLOS Genetics. Congratulations!

I evaluated your responses to the reviews received through Review Commons (for full transparency, in doing so the system allows me to see the identities of the reviewers).  I found the reviewers' comments to be careful and scholarly and your responses/revisions to be equally thoughtful, and to have sufficiently addressed their critiques, and I don't feel the need for further external evaluation (again, for full transparency, this is not always the case with post-revision Review Commons submissions). 

Yours sincerely,

Gregory P. Copenhaver

Editor-in-Chief

PLOS Genetics

Comments from the reviewers (if applicable):

**Data Deposition**

http://datadryad.org/submit?journalID=pgenetics&manu=PGENETICS-D-20-01814R1

**Press Queries**

---

## [Editor Report · Acceptance letter]

21 Mar 2021

PGENETICS-D-20-01814R1 

The Capacity of Origins to Load MCM Establishes Replication Timing Patterns 

Dear Dr Rhind, 

We are pleased to inform you that your manuscript entitled "The Capacity of Origins to Load MCM Establishes Replication Timing Patterns" has been formally accepted for publication in PLOS Genetics! Your manuscript is now with our production department and you will be notified of the publication date in due course.

With kind regards,

Alice Ellingham

PLOS Genetics

On behalf of:
